# A High-Functional-Density Integrated Inertial Switch for Super-Quick Initiation and Reliable Self-Destruction of a Small-Caliber Projectile Fuze

**DOI:** 10.3390/mi14071377

**Published:** 2023-07-05

**Authors:** Bo He, Yong Yuan, Jie Ren, Wenzhong Lou, Hengzhen Feng, Mingrong Zhang, Sining Lv, Wenting Su

**Affiliations:** 1School of Mechatronical Engineering, Beijing Institute of Technology, Beijing 100081, China; cqrhb0928@126.com (B.H.);; 2Science and Technology on Electromechanical Dynamic Control Laboratory, Beijing Institute of Technology, Beijing 100081, China; 3Chongqing Innovation Center, Beijing Institute of Technology, Chongqing 401120, China; 4Chongqing Changan Wangjiang Industrial Group Co., Ltd., Chongqing 401120, Chinazhangmr@163.com (M.Z.)

**Keywords:** small-caliber projectile fuze, super-quick initiation, reliable self-destruction, high-functional-density integrated, inertial switch

## Abstract

With the aim of achieving the combat technical requirements of super-quick (SQ) initiation and reliable self-destruction (SD) of a small-caliber projectile fuze, this paper describes a high-functional-density integrated (HFDI) inertial switch based on the “ON-OFF” state transition (i.e., almost no terminal ballistic motion). The reliable state switching of the HFDI inertial switch is studied via elastic–plastic mechanics and verified via both simulations and experiments. The theoretical and simulation results indicate that the designed switch can achieve the “OFF-ON” state transition in the internal ballistic system, and the switch can achieve the “ON-OFF” state transition in the simulated terminal ballistic system within 8 μs or complete the “ON-OFF” state transition as the rotary speed sharply decreases. The experimental results based on the anti-target method show the switch achieves the “ON-OFF” state transition on the μs scale, which is consistent with the simulation results. Compared with the switches currently used in small-caliber projectile fuzes, the HFDI inertial switch integrates more functions and reduces the height by about 44%.

## 1. Introduction

Small-caliber artillery ammunition is among the most consumed weaponry in practical training. The high initiation reliability of fuzes directly improves the damage effectiveness of ammunition and reduces the number of dangerous duds during training, which is a critical combat technical indicator [1]. In a small-caliber projectile fuzing system, the inertial components have always been the core means by which the fuze recognizes changes in the ballistic environment [2]. Especially in an electromechanical fuzing system, the inertial switch is the key to the reliable response of the fuze to the impact overload of the projectile hitting the target. The typical conditions of a small-caliber projectile fuze hitting a 2 mm thick aluminum alloy plate are listed in Table 1.

Research on improvement of inertial switches applied to fuzes has been reported at recent sessions of the American Fuze Annual Conference. This involves dividing the sensitive structures of inertial switches into spring–mass and cantilever beam types [3,4,5,6].

In recent years, the research focus on spring–mass inertial switches has evolved from traditional coil-spring–mass structures to planar spring–mass structures based on micro-electro-mechanical systems (MEMS) technology [7,8,9,10,11]. Inertial switches based on traditional coil spring–mass structures have a strong ability to withstand highly dynamic environments (a maximum axial overload not less than 70,000 g and maximum rotary speed not less than 70,000 rpm), which are typical working environments for small-caliber projectile fuzes. However, the traditional coil spring has a large volume that is difficult to fit within the limited internal space of the fuze. Compared with traditional coil spring–mass structures, planar spring–mass structures based on MEMS technology are more integrated and can match the limited space of a small-caliber projectile fuze. However, owing to the low yield rate of plane springs and the large difference between the actual and the theoretical mechanical properties of plane springs in a highly dynamic environment with an acceleration greater than the order of 10^3^ g, no inertial switch with plane spring–mass structures has been applied to small-caliber projectile fuzes.

Cantilever beam structures, as typical inertial overload-sensitive structures, have been widely used in the design of high g-value accelerometers and have shown excellent performance in withstanding highly dynamic environments [12,13,14]. Ning et al. [15] designed an inertial switch based on a cantilever beam structure to respond to high axial overload in the direction of the projectile axis. When the fuze hit the target, the cantilever beam structure bent in response to forward impact overload and was electrically connected to the fixed electrode to achieve a switch state transition. Compared to spring–mass structures, cantilever beam structures are simple to manufacture. The inertial switch can also withstand highly dynamic environments while having a small overall volume, which gives it great potential for application in small-caliber projectile fuzes.

It must be noted that the small-caliber projectile fuze has the combat technical requirements of super-quick (SQ) initiation and reliable self-destruction (SD) [16]. However, existing inertial switches are based on the “OFF-ON” state transition rather than the “ON-OFF” state transition, which results in longer response time (ms level). At present, an inertial switch has not been seen that can quickly respond to axial overload on the basis of the “ON-OFF” state transition or respond to the rotational change of the projectile, which helps the fuze achieve reliable SD based on the attenuation in projectile rotary speed.

With the aim of achieving the combat technical requirements of SQ initiation and reliable SD for a small-caliber projectile fuze, a high-functional-density (HFDI) inertial switch based on the “ON-OFF” state transition is described in this paper. First, the model and working logic of the switch are designed on the basis of the axial/radial overload variation of the whole small-caliber projectile fuze. Second, a computer-aided design (CAD) model of the switch is established, and LS-DYNA software is used to perform explicit dynamics simulations based on the whole ballistic inertia overload. The simulation results show the correctness of the working principle of the switch. The switch is then processed and assembled. Finally, the switch is tested using a home-built highly dynamic testing system based on the anti-target method.

## 2. Design and Principle: The HFDI Inertial Switch

### 2.1. Structural Design

The designed HFDI inertial switch model is shown in Figure 1. The switch is placed on the fuze control circuit board (11) in the vertical direction of the rotary axis and at a certain distance from the rotary axis.

In the initial state, the axially sensitive layers (2-1, 2-2) and GND layers (6-1, 6-2), which consist of conductive material, are not interconnected with each other. After the projectile is launched, the axially sensitive layers respond to axial setback overload. The structure of axially sensitive layer #1 is a cantilever beam mass with a lower natural frequency. Its stronger response under setback overload helps the cantilever beam with the higher natural frequency on axially sensitive layer #2 to generate elastic–plastic deformation. As the setback overload increases, the cantilever beam on axially sensitive layer #2 contacts the copper foil (4) below insulating layer #1 (3-1). The copper foil is initially connected to the GND layers. Because the substrate (5) of the copper foil consists of flexible material (i.e., PDMS), the cantilever beam on axially sensitive layer #2 continues to generate greater elastic–plastic deformation after contact with the copper foil. As the setback overload gradually decreases and disappears, axially sensitive layer #1 releases axially sensitive layer #2. The cantilever beam of axially sensitive layer #2 generates elastic recovery due to the strain characteristics of the elastic–plastic material, but the recovery deformation of the flexible substrate can ensure that the copper foil is always stably connected to the cantilever beam on axially sensitive layer #2 (i.e., the ON state). Furthermore, owing to the high natural frequency of the cantilever beam on axially sensitive layer #2, this layer is insensitive to low-amplitude axial overload in the external ballistic system, including air damping and possible impact on raindrops. In the terminal ballistic system, the cantilever beam of axially sensitive layer #2 generates elastic–plastic deformation in response to large axial overload, resulting in disconnection from the copper foil (i.e., the “OFF” state).

In the initial state, the radially sensitive layer (7) and GND layers, which consist of conductive material, are not interconnected with each other. After the projectile is launched, the radially sensitive layer responds to radial centrifugal overload. As the rotary speed of the projectile increases, the deformation of the sensitive structure based on the Y-shaped cantilever beam gradually increases until it contacts the circular boss of GND layer #2. The stable centrifugal force provided by the projectile rotation is sufficient to support a reliable connection between the radially sensitive layer and GND layer before the fuze hits the target (i.e., the “ON” state). In the terminal ballistic system, when the radial overload of the Y-shaped cantilever beam response is greater than the centrifugal force in the opposite direction, or when the centrifugal force is insufficient to resist the elastic recovery force of the Y-shaped cantilever beam with the attenuation in projectile rotary speed, the Y-shaped cantilever beam of the radially sensitive layer and the circular boss of GND layer #2 are disconnected (i.e., the “OFF” state).

### 2.2. Working Logic

The working logic of the fuze based on the HFDI inertial switch is shown in Figure 2. When the fuze is launched with the projectile and enters the internal ballistic environment, the switch responds to both setback overload and centrifugal overload in highly dynamic environments, and the cantilever beam structures of the axially/radially sensitive layers move. When the fuze is armed through software timing or a clock mechanism, the fuze microcontroller detects the levels of the two I/O ports, which are connected to the switch. According to the peripheral circuit in Figure 1, the levels of the two I/O ports under normal conditions should be low, indicating that the axially/radially sensitive layers of the switch have already moved into place. Then, the fuze microcontroller activates the rising-edge-interrupt-triggering function of the corresponding I/O ports and will continuously detect the level changes of the I/O ports at a frequency of 1 MHz.

When the fuze hits the target at different incident angles (i.e., angles between the axis of the projectile and the normal direction of the target surface), as shown in Table 1, the axial/radial overload increases/decreases with the increase in incident angle, respectively. Under different terminal ballistic conditions, the three situations for the state transition of the switch are as follows:(i).When the fuze hits the target at a small incident angle, the axial component of the impact overload is large. The switch achieves the “ON-OFF” state transition through the axially sensitive structural response.(ii).When the fuze hits the target at a large incident angle, the radial component of the impact overload is large, and the switch achieves the “ON-OFF” state transition through the radially sensitive structure’s response. However, it should be noted that owing to the rotation of the projectile, the direction of radial overload when the fuze hits the target is unpredictable. When the direction of overload is similar to the direction of centrifugal force on the radially sensitive structure, the switch cannot reliably complete the state transition.(iii).When the fuze misses the target and flies for a longer distance, there is no axial/radial overload. The centrifugal force on the radially sensitive structure decreases with the attenuation in projectile rotary speed and is insufficient to resist the restoring force of material elastic deformation. The switch achieves the “ON-OFF” state transition through elastic recovery of the axially sensitive structure.

When the fuze microcontroller detects the change in the rising edge of the corresponding I/O ports, it outputs the initiation signal to achieve SQ detonation of the warhead or reliable SD.

### 2.3. Theoretical Analysis

The designs of the axially/radially sensitive structures, which are the core structures of the switch, are shown in Figure 3. The key to achieving the “ON-OFF” state transition of the switch in the terminal ballistic system is to switch from “OFF” to “ON” in the internal ballistic system and maintain stability in the external ballistic system.

Assuming that the cantilever beam structures of the axially/radially sensitive layers are elastic–plastic beam structures, the deflection of the beam structures in the internal ballistic system is analyzed using the beam deflection equation in material mechanics. The deflection of the cantilever beam structures of axially/radially sensitive layers is calculated using
(1)$$w_{x}=\frac{Fl^{2}}{6E\left(\varepsilon \right)I}\left(l-3x\right)$$
where F is the setback/centrifugal inertial force, l is the distance between the centroid of the cantilever beam structure and the fixed end, $E\left(\varepsilon \right)$ is the change in the Young modulus of the material with strain, and I is the moment of inertia of the section.

The natural frequency of the cantilever beam–mass structure in axially sensitive layer #1 is much lower than that of the three cantilever beams in axially sensitive layer #2 (the two in the middle are buffer cantilever beams, and the one on the right is a connecting cantilever beam). Assuming that the cantilever beam–mass structure of axially sensitive layer #1 is the free mass, the inertial force generated under setback overload is uniformly distributed on the three cantilever beams of axially sensitive layer #2. The deflection of the free end of the connecting cantilever beam over time can be written as
(2)$$w_{a}\left(t\right)=-\frac{16F_{S}\left(t\right)L2_{a}^{3}+15F_{b}\left(t\right)L2_{a}^{3}}{144E\left(\varepsilon \right)I_{a}}$$
where $F_{S}\left(t\right)$ is the setback inertial force of the cantilever beam–mass structure of axially sensitive layer #1, $F_{b}\left(t\right)$ is the setback inertial force of the connecting cantilever beam of axially sensitive layer #2, and $I_{a}$ is the moment of inertia of the section of the connecting cantilever beam. The equations for $F_{S}\left(t\right)$, $F_{b}\left(t\right)$, and $I_{a}$ are
(3)$$F_{S}\left(t\right)=\rho \left(\left(L1_{a}-R_{a}\right)W1_{a}+\pi R_{a}^{2}\right)H1_{a}a\left(t\right)$$
(4)$$F_{b}\left(t\right)=\rho L2_{a}W2_{a}H2_{a}a\left(t\right)$$
(5)$$I_{a}=\frac{W2_{a}H2_{a}^{3}}{12}$$

The cantilever beam structure of the radially sensitive layer responds to radial centrifugal overload. The deflection of the free end of the cantilever beam over time can be written as
(6)$$w_{r}\left(t\right)=-\frac{F_{c}\left(t\right)l_{\mathrm{mass}}^{2}}{6E\left(\varepsilon \right)I_{r}}\left(3L1_{r}+3L2_{r}+3L3_{r}-l_{\mathrm{mass}}\right)$$
where $l_{\mathrm{mass}}$ is the centroid of the cantilever beam of the radially sensitive layer, $F_{c}\left(t\right)$ is the centrifugal inertial force of the cantilever beam of the radially sensitive layer, and $I_{r}$ is the moment of inertia of the section of the cantilever beam. The equations for $F_{c}\left(t\right)$ and $I_{r}$ are
(7)$$F_{c}\left(t\right)=\rho \left(L1_{r}W1_{r}+L2_{r}W2_{r}+2L3_{r}W3_{r}\right)H_{r}\omega^{2}\left(t\right)R$$
(8)$$I_{r}=\frac{H_{r}W1_{r}^{3}}{12}$$

According to the structural design in Section 2.1, before the setback overload reaches its peak, the connecting cantilever beam of axially sensitive layer #2 contacts the copper foil (i.e., $w_{a}$ > $H_{3-1}$, where $H_{3-1}$ is the thickness of insulating layer #1). Before the end of the internal ballistic system, the cantilever beam of the radially sensitive layer contacts the circular boss of GND layer #2 (i.e., $w_{r}$ > $W2_{r}/2-W3_{r}-R_{f}$).

The conductive material is brass, with a density of 8912.9 kg/m^3^ and Young modulus of 117.2 GPa. The main structural parameters of the switch are listed in Table 2. According to modal analysis, the natural frequencies of the cantilever beam–mass structure of axially sensitive layer #1 and the cantilever beam of axially sensitive layer# 2 are 12.656 Hz and 112.66 Hz, respectively. Figure 4 shows the theoretical results for the axially/radially sensitive layers responding to the internal ballistic environment of the small-caliber projectile fuze. Before the setback overload reaches its peak, $w_{a}=110$ μm > $H_{3-1}=100$ μm, and before the end of the internal ballistic system, $w_{r}=160$ μm > $W2_{r}/2-W3_{r}-R_{f}=150$ μm. This means that the designed switch can achieve the “OFF-ON” state transition in the internal ballistic environment.

## 3. Dynamic Simulations of the Whole Ballistic System

In order to improve simulation efficiency, the axially/radially sensitive components of the switch were modeled and simulated based on the whole ballistic inertial overload.

### 3.1. Dynamic Simulations of Axial Overload

Figure 5 shows the finite element model of axially sensitive components. *CONTACT_AUTOMATIC_NODES_TO_SURFACE is used to define the contact relationship between axial sensitive layer #2 and other components, while *CONTACT_AUTOMATIC_SINGLE_SURFACE is used to define the contact relationship between other components. *BOUNDARY_PRESCRIBED_MOTION_NODE is used to load the model with axial overload. The compositions of the axially sensitive components and key material parameters are listed in Table 3. The distance between node1 on the connecting beam and node2 on the surface of the copper foil ($D_{S}$) is selected as the key parameter. Figure 6 shows the variational trend of $D_{S}$ with and without cushioning beams based on the simulated whole ballistic axial overload. The full ballistic axial overload is composed of the internal/external ballistic axial overload calculated using an empirical formula and a triangular pulse for simulating terminal ballistic axial overload with a high peak and short duration. The cushioning beams not only effectively slow down the compression deformation by the upper beam mass of the connecting beam in the early stage of the internal ballistic process, which can prevent an accidental drop from causing deformation of the connecting beam, but also prevent the connecting beam from being excessively compressed by the upper beam mass.

Figure 7 shows the Von Mises stress cloud of axially sensitive components. At 0.5 ms, under the action of the setback inertial force ($F_{S}$), the connecting beam touches the copper foil and begins to compress the PDMS. The elastic recovery force ($F_{T}$) of the PDMS helps the connecting beam resist deformation. At 1 ms, the values of $F_{S}$ and $F_{T}$ reach their maxima. As the setback overload decreases, $F_{S}$ is less than $F_{T}$, and the PDMS begins to recover from deformation. The deformation process offsets the elastic recovery deformation of the connecting beam, which means that the connecting beam and copper foil supported by the PDMS maintain a reliable electrical connection at the ends of the internal ballistic and external ballistic environments. At 4.5 ms, a forward impact overload with an amplitude of 4000 g was loaded to simulate the impact of a large-diameter raindrop on the projectile during flight. It can be seen that the connecting beam and copper foil still maintain a stable connection, which verifies that the connecting beam is not sensitive to low-peak impact overload because of its low natural frequency. Using a forward impact overload with an amplitude of 130,000 g and pulse width of 10 μs in the simulated terminal ballistic process, the connecting beam and copper foil are disconnected within 8 μs, which helps the small-caliber projectile fuze achieve the SQ initiation function.

### 3.2. Dynamic Simulations of Radial Overload

Figure 8 shows the finite element model of radially sensitive components. *CONTACT_AUTOMATIC_SINGLE_SURFACE is used to define the contact relationship between all components. *BOUNDARY_PRESCRIBED_MOTION_NODE is used to load the model with rotary speed. The compositions of the radially sensitive components and key material parameters are listed in Table 4. The distance between node1 on the circular boss of the GND layer and node2 on the cantilever beam of the radially sensitive layer ($D_{C}$) is selected as the key parameter. The variational trends of $D_{C}$ with different eccentric distances (R) based on the simulated whole ballistic radial overload are shown in Figure 9. The response speed of the radially sensitive components in the internal ballistic system is faster farther away from the projectile axis. When R=4 mm, the radially sensitive components cannot transition to the “ON” state in the internal ballistic process. When R is greater than 5 mm, the radially sensitive components can achieve the switch state transition in the internal ballistic system and maintain the “ON” state in the external ballistic system. In the simulated terminal ballistic process, as the rotary speed sharply decreases, the radially sensitive components switch from “ON” to “OFF”.

Figure 10 shows the Von Mises stress cloud of the radially sensitive components when R=6 mm. In the internal ballistic system, the radially sensitive components change from the “OFF” state to the “ON” state (Figure 10a). In the external ballistic system, the radially sensitive components remain in a stable “ON” state (Figure 10b). In the terminal ballistic system, the radially sensitive components achieve the “ON-OFF” state transition through two operating conditions, as described in Section 2.2. Figure 10c shows that when the rotary speed decreases, the radially sensitive components achieve the “ON-OFF” state transition, which helps the small-caliber projectile fuze achieve reliable SD function.

## 4. Highly Dynamic Testing

Figure 11 shows the finished products of each layer of the designed HFDI inertial switch, as well as the complete switch obtained through assembly. All layers are manufactured using picosecond ultraviolet laser technology, with a positioning accuracy higher than 5 um and a minimum machining feature size of 10 μm. Using 3D printing to manufacture assembly fixtures with shapes complementary to the designed HFDI inertial switch, each layer is placed in the assembly fixture in sequence, and finally, all layers are fixed with insulated bolts. The overall size of the designed HFDI inertial switch is 5.2×5.2×2.8 mm^3^ (including a 1 mm thick screwhead), which reduces the height of the switch by about 44% compared with the currently used small-caliber projectile fuze.

A home-built highly dynamic testing system was used to manufacture the “high peak, narrow pulse” impact component of the terminal ballistic system of the small-caliber projectile fuze for switch state transition testing. The composition of the home-built highly dynamic testing system is shown in Figure 12.

As illustrated by the system diagram in Figure 12, the testing system was built on the basis of the anti-target method. The light-gas gun utilized high-pressure gas to accelerate the sabot target, and the target was shot into the recycling bin. The target impacted with high speed the equivalent fuze carrier suspended on the fixed support. The designed HFDI inertial switch was fixed inside the equivalent fuze carrier. By using lead wires, external power was supplied to the switch testing circuit, and the oscilloscope was used to collect the switch state transition signal. A high-speed camera was used to observe the speed at which the target entered. Compared with a forward-firing testing system, the home-built testing system based on the anti-target method has advantages such as easy real-time signal acquisition, easy adjustment of fuze attitude, and low time cost.

Figure 13 shows the equivalent fuze carrier, switch testing circuit, and electrical connections. Owing to the inability to simulate the inertial environment of the internal ballistic system of the small-caliber projectile fuze in the laboratory, the connecting beam of axially sensitive layer #2 of the switch was manually deformed to the “ON” state by gently pressing it with tweezers before testing. A 3.3 V DC power supply was provided through the power interface. According to the peripheral circuit design, the oscilloscope collected the stable low-level signal from the test interface. Figure 14 shows the level change of the oscilloscope through the rising-edge-triggering acquisition to the test interface when the target (solid wood) impacts the equivalent fuze carrier at a speed of about 300 m/s. The test interface level achieved the low-to-high transition on the μs scale, which indicated that the switch could achieve rapid response to terminal ballistic impact. This verified the correctness of the simulations. The switch remained stable at high levels for about 50 μs. This meant that the fuze microcontroller could perform a short delay to confirm the I/O level after detecting the rising edge, in order to eliminate the risk of interference signals.

## 5. Conclusions

In this paper, we have described how we designed an HFDI inertial switch to achieve the combat technical requirements of SQ initiation and reliable SD for a small-caliber projectile fuze. First, the structure and whole ballistic working logic of the switch were designed, and the axially/radially sensitive components of the switch were theoretically analyzed on the basis of elastic–plastic mechanics. The theoretical results indicate that the designed switch could achieve the “OFF-ON” state transition in the internal ballistic environment. Second, a dynamic simulation analysis of the whole ballistic process of the proposed device was carried out using LS-DYNA software. The simulation results showed that the switch could achieve the “ON-OFF” state transition in the simulated terminal ballistic system within 8 μs and complete the “ON-OFF” state transition as the rotary speed sharply decreased. This validated the working logic of the designed switch. Subsequently, the designed switch was processed and assembled. Compared with the switches currently used in small-caliber projectile fuzes, the HFDI inertial switch integrates more functions and reduces the height by about 44%. Finally, the switch was tested using a home-built highly dynamic testing system based on the anti-target method. When the target collided with the equivalent fuze carrier (with a switch inside) at a relative speed of about 300 m/s, the switch that was already in the “ON” state could achieve the “ON-OFF” state transition on the μs scale, which was consistent with the simulation results.

Compared to the “OFF-ON” state transition, the “ON-OFF” state transition in the terminal ballistic system has a shorter response time. Combined with the rapid attenuation in projectile rotary speed at any ballistic endpoint, this indicates that the HFDI inertial switch can help a small-caliber projectile fuze achieve SQ initiation and reliable SD.

## Figures and Tables

**Figure 1 micromachines-14-01377-f001:**
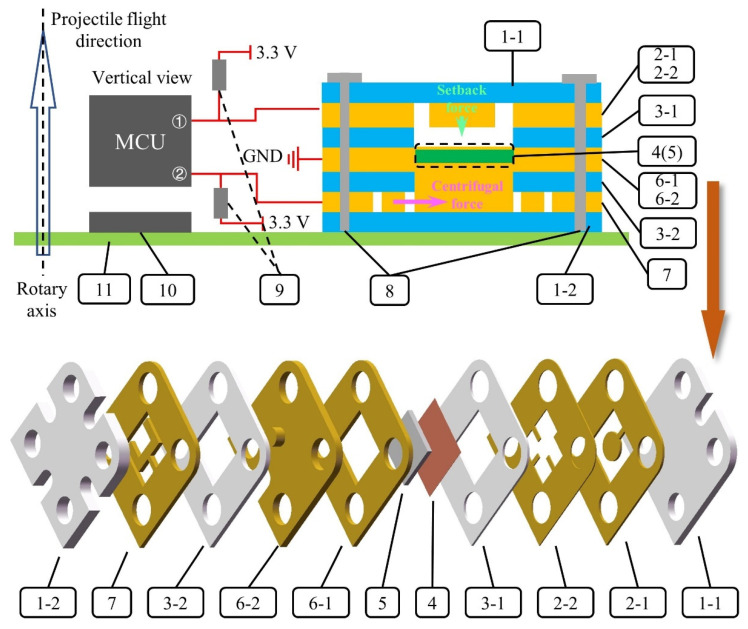
Designed HFDI inertial switch model. 1-1: upper cover plate; 1-2: bottom cover plate; 2-1: axially sensitive layer #1; 2-2: axially sensitive layer #2; 3-1: insulating layer #1; 3-2: insulating layer #2; 4: copper foil; 5: flexible substrate; 6-1: GND layer #1; 6-2: GND layer #2; 7: radially sensitive layer; 8: insulating bolt; 9: pull-up resistance; 10: fuze microcontroller; 11: PCB.

**Figure 2 micromachines-14-01377-f002:**
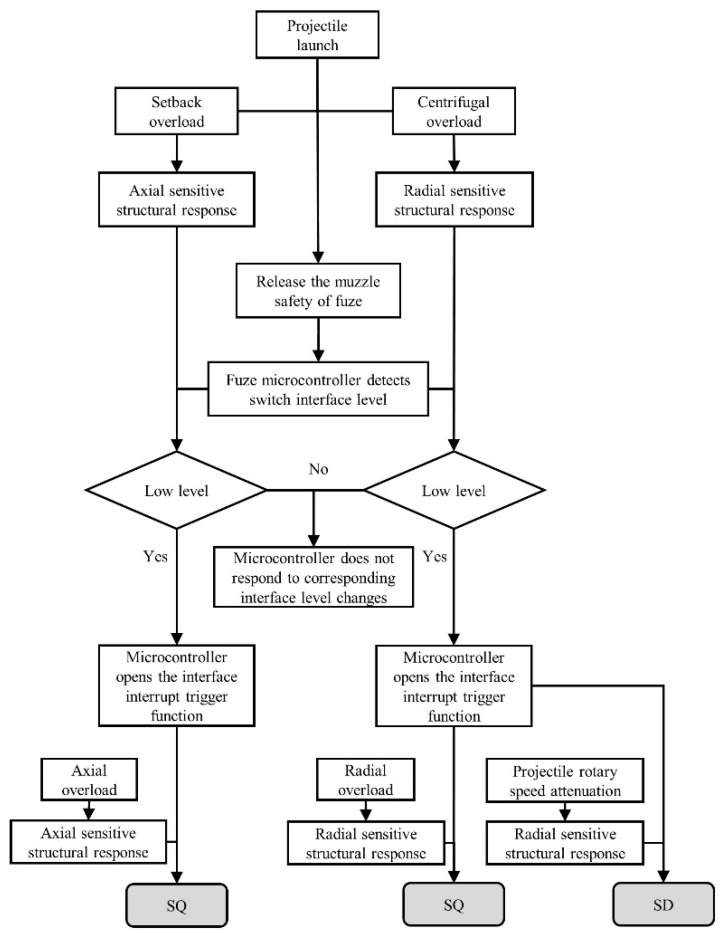
Working logic of the fuze based on the HFDI inertial switch.

**Figure 3 micromachines-14-01377-f003:**
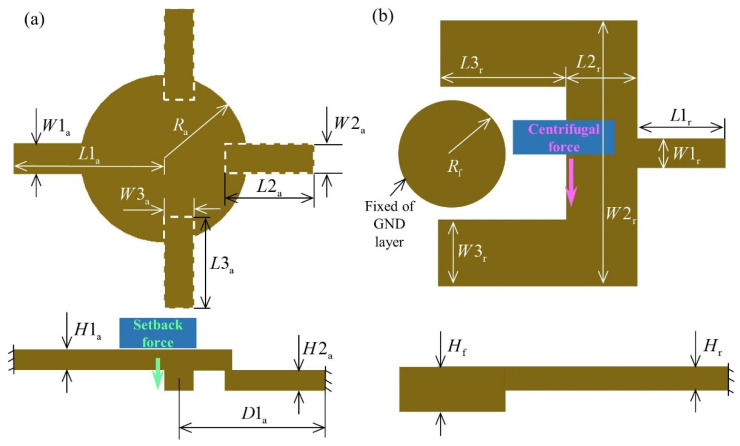
Design scheme of the core structures of the switch. (**a**) Axially sensitive structure; (**b**) radially sensitive structure.

**Figure 4 micromachines-14-01377-f004:**
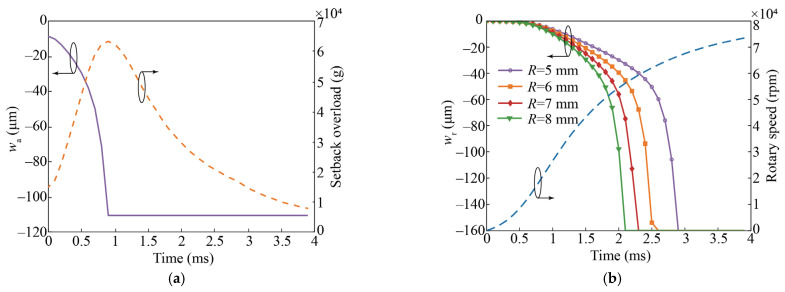
Theoretical results for the response process of the designed switch in the internal ballistic environment. (**a**) Axially sensitive structure; (**b**) radially sensitive structure.

**Figure 5 micromachines-14-01377-f005:**
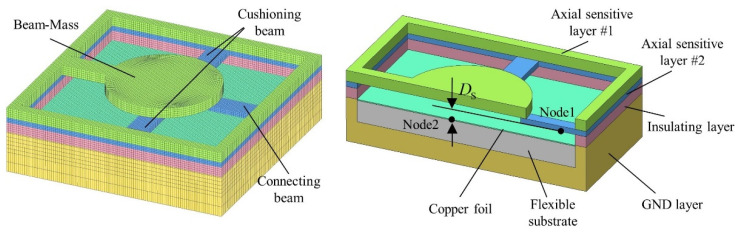
Finite element model of axially sensitive components.

**Figure 6 micromachines-14-01377-f006:**
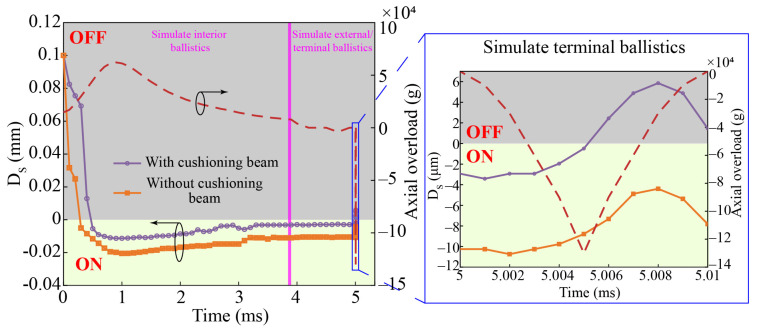
Variational trend of $D_{S}$ with and without a cushioning beam based on the simulated whole ballistic axial overload.

**Figure 7 micromachines-14-01377-f007:**
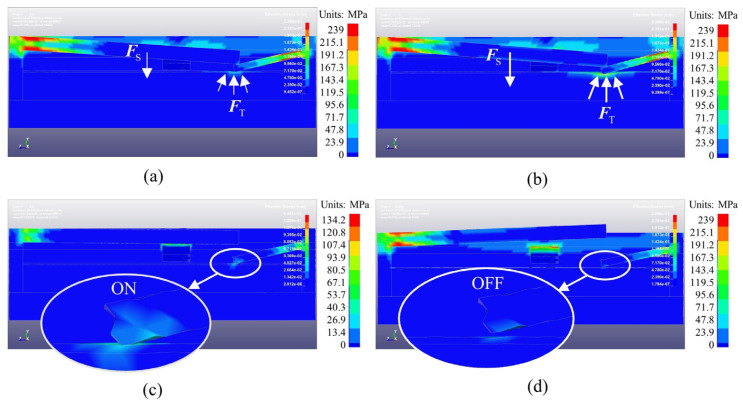
Von Mises stress cloud of axially sensitive components. (**a**) 0.5 ms; (**b**) 1 ms; (**c**) 4.5 ms; (**d**) 5.008 ms.

**Figure 8 micromachines-14-01377-f008:**
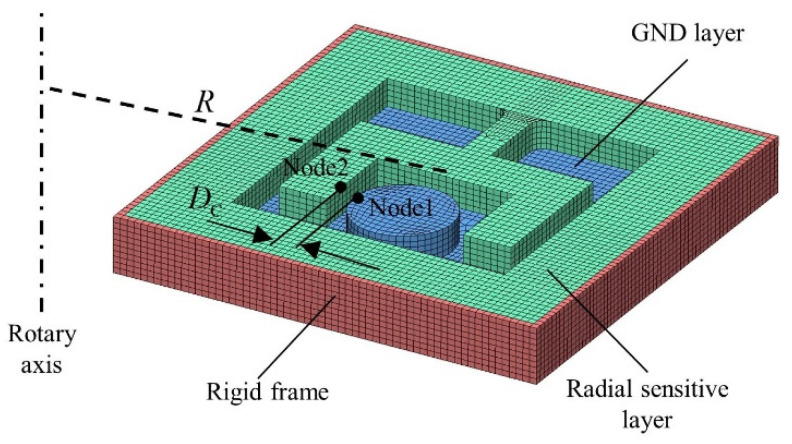
Finite element model of radially sensitive components.

**Figure 9 micromachines-14-01377-f009:**
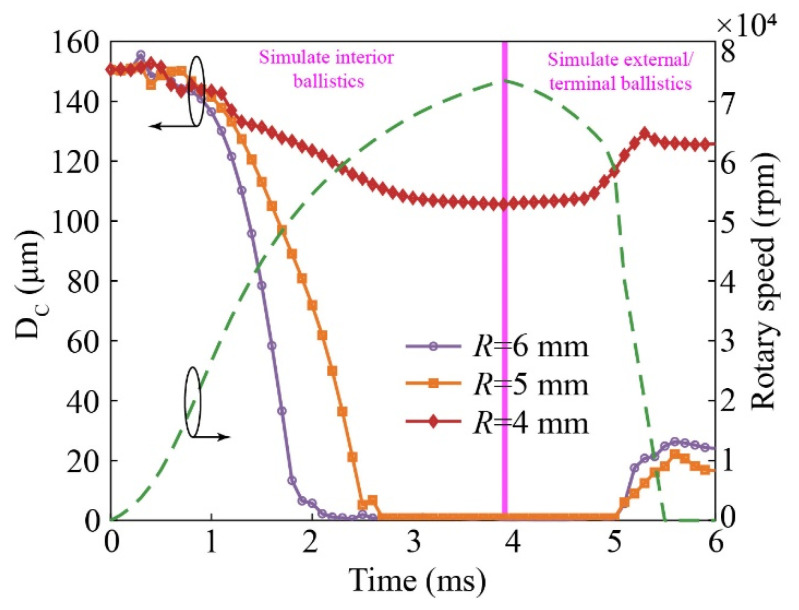
Variational trends of $D_{S}$ with different eccentric distances based on the simulated whole ballistic radial overload.

**Figure 10 micromachines-14-01377-f010:**
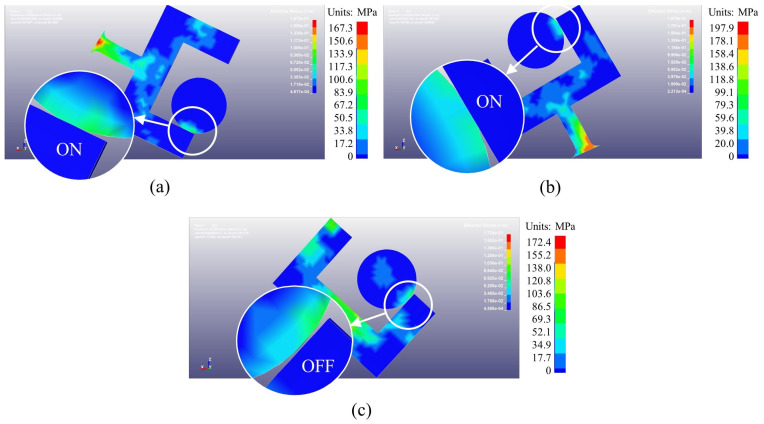
Von Mises stress cloud of radially sensitive components. (**a**) 2.3 ms; (**b**) 4.4 ms; (**c**) 5.2 ms.

**Figure 11 micromachines-14-01377-f011:**
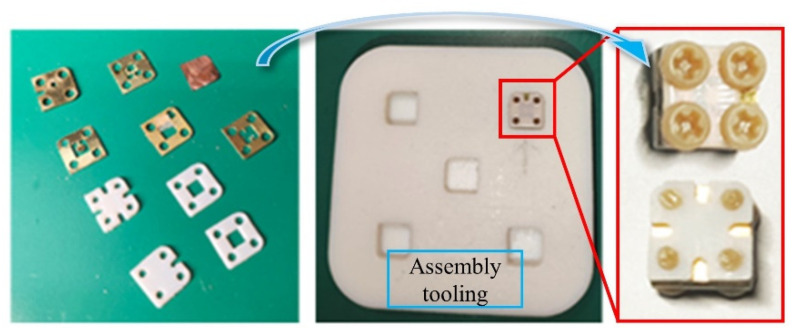
Finished product model of the designed HFDI inertial switch.

**Figure 12 micromachines-14-01377-f012:**
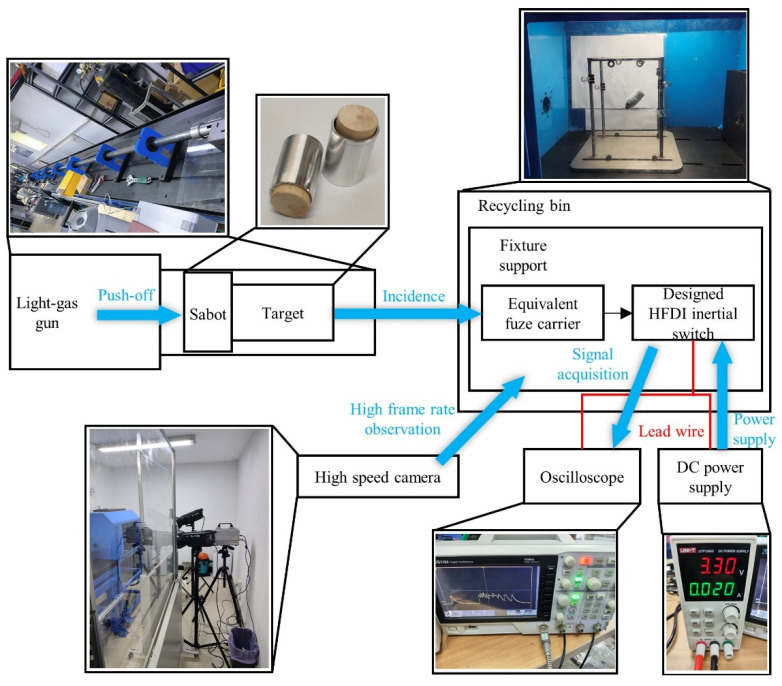
Home-built highly dynamic testing system based on the anti-target method.

**Figure 13 micromachines-14-01377-f013:**
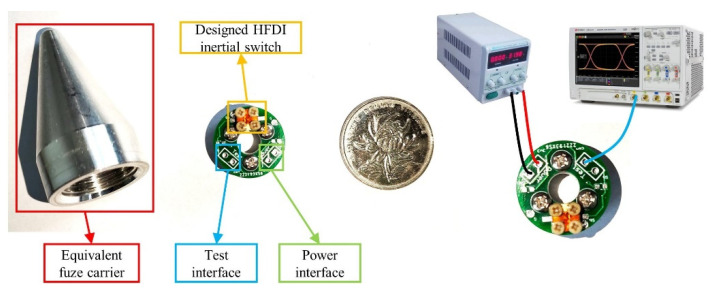
Equivalent fuze carrier, switch testing circuit, and electrical connections.

**Figure 14 micromachines-14-01377-f014:**
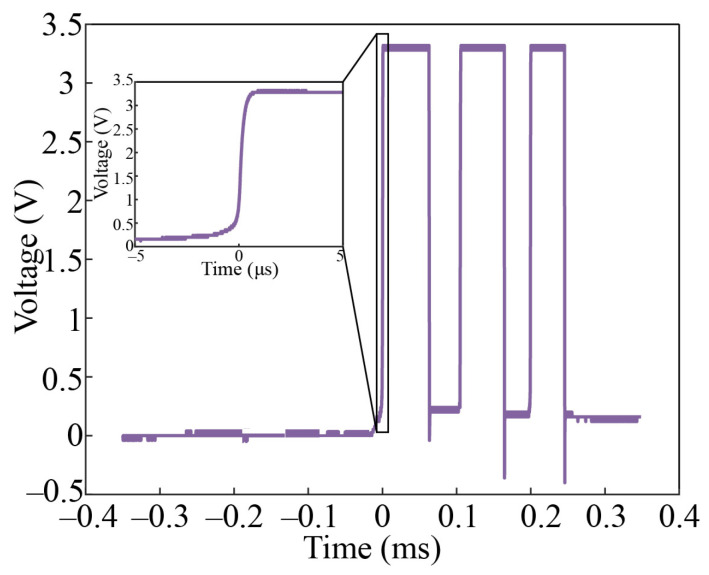
Real-time collection of the test signal.

**Table 1 micromachines-14-01377-t001:** Typical conditions of a small-caliber projectile fuze hitting a 2 mm thick aluminum alloy plate.

Caliber (mm)	Flight Distance (m)	Speed (m/s)	Rotary Speed (rpm)	Incident Angle (°)	Axial/Radial Overload Peak (g)	Axial/Radial Overload Pulse Width (μs)
30	2000	430	66,000	0	180,000/≈0	7/≈0
				30	140,000/12,000	7/>20
				60	60,000/24,000	8/>20
				90	10,000/26,000	9/>20

**Table 2 micromachines-14-01377-t002:** Main geometric parameters of the designed HFDI inertial switch.

Components	Geometric Parameters	Value (μm)	Components	Geometric Parameters	Value (μm)
1-1: Upper cover plate	$\mathrm{Thickness}\;H_{1-1}$	200	4: Copper foil	$\mathrm{Thickness}\;H_{4}$	10
1-2: Bottom cover plate	$\mathrm{Thickness}\;H_{1-2}$	200	5: Flexible substrate	$\mathrm{Thickness}\;H_{5}$	200
2-1: Axial sensitive layer #1	$\mathrm{Radius}\;R_{a}$	550	6-1: GND layer #1	$\mathrm{Thickness}\;H_{6-1}$	200
	$\mathrm{Width}\;W1_{a}$	200	6-2: GND layer #2	$\mathrm{Thickness}\;H_{6-2}$	700
	$\mathrm{Length}\;L1_{a}$	1000		$\mathrm{Thickness}\;H_{f}$	400
	$\mathrm{Thickness}\;H1_{a}$	100		$\mathrm{Radius}\;R_{f}$	350
2-2: Axial sensitive layer #2	$\mathrm{Width}\;W2_{a}$	200	7: Radial sensitive layer	$\mathrm{Width}\;W1_{r}$	200
	$\mathrm{Length}\;L2_{a}$	500		$\mathrm{Length}\;L1_{r}$	600
	$\mathrm{Width}\;W3_{a}$	200		$\mathrm{Width}\;W2_{r}$	1700
	$\mathrm{Length}\;L3_{a}$	600		$\mathrm{Length}\;L2_{r}$	400
	$\mathrm{Distance}\;D1_{a}$	900		$\mathrm{Width}\;W3_{r}$	350
	$\mathrm{Thickness}\;H2_{a}$	50		$\mathrm{Length}\;L3_{r}$	800
3-1: Insulating layer #1	$\mathrm{Thickness}\;H_{3-1}$	100		$\mathrm{Thickness}\;H_{r}$	200
3-2: Insulating layer #2	$\mathrm{Thickness}\;H_{3-2}$	200			

**Table 3 micromachines-14-01377-t003:** Compositions of axially sensitive components and key material parameters.

Components	Materials	Constitutive Model	Young’s Modulus (GPa)	Density (kg·m^−3^)	Poisson’s Ratio	Yield Strength (MPa)
Axial sensitive layer #1	Brass H62 [17]	MAT_PLASTIC_KINEMATIC	117.2	8912.9	0.33	239
Axial sensitive layer #1						
GND layer						
Insulating layer	Nylon [18]	MAT_PLASTIC_KINEMATIC	4.5	1100.0	0.375	98
Copper foil	Copper foil [19]	MAT_PLASTIC_KINEMATIC	115	7930.0	0.33	195
Flexible substrate	PDMS	MAT_PLASTIC_KINEMATIC	0.003	970.0	0.49	–

**Table 4 micromachines-14-01377-t004:** Compositions of radially sensitive components and key material parameters.

Components	Materials	Constitutive Model	Young’s Modulus (GPa)	Density (kg·m^−3^)	Poisson’s Ratio	Yield Strength (MPa)
Radial sensitive layer #1	Brass H62 [17]	MAT_PLASTIC_KINEMATIC	117.2	8912.9	0.33	239
GND layer						

## Data Availability

The data presented in this study are available on request from the corresponding author.

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
