# Peer review of "A High-Functional-Density Integrated Inertial Switch for Super-Quick Initiation and Reliable Self-Destruction of a Small-Caliber Projectile Fuze"

_micromachines, 2023, doi:10.3390/mi14071377_

Round 1

Reviewer 1 Report

Dear authors,

first many thanks for your contribution. It's quite interesting topic and use-case.

It's difficult to read and follow your text. May be it's a good idea to re-write the introduction to provide readers more background informations.

It's also important to know the boundary conditions or requirements for the use-case - like impact, acceleration, speed, temperature etc. and the state-of-the Art in use yet. Which systems or devices are known?

Page 1-DUD? what does it mean?
-typical conditions in the use-case, impact, acceleration, speed, temperature etc. in a table
Page 2-3
-sensitive layer #2 - plastic properties?
Page 4
-the critical angle 60°. More information and explaination why the 3 situations happen at critical angle of 60°?
-And the 3. situation - what does it mean neither >60 nor =<60°?. Which angle is meant here?
-Also not clear the response of the axial sensitive structure after it has been plastic deformed in the lauch phase
already. The OFF-ON is quite clear but not the ON-OFF behavior.
Page 5
-How large are the natural frequencies of the cantiveler beams in this work?
Page 7
-we have here the contact mechanics issue - how it has been solved in this simulation work?
-The model for terminal ballistics has to be explained how the OFF-ON behavior is.
Page 8-9
-The same issue with radial sensitive components
Page 10-11
-Explain the manufaturing process and the realized single parts (tolerances etc.)
-explain the assembling and electrical contacting the device

BR

It's hard to understand and follow. May be it's a good idea to use the editing service from MDPI.

Reviewer 2 Report

In this paper, a HFDI inertia switch was designed to achieve the combat technical requirements of SQ initiation and reliable SD of small caliber projectile fuze. This work is useful for related miniaturization devices. However, there are some comments should be addressed, as follows:

1.      Some abbreviations should be indicated when they appeared firstly in the manuscript content, such as, MEMS, Etc.

2.      On page 2, Line 56, the reference [15] from Ning et al was cited incorrectly, which is not corresponding to the references at the end of the manuscript. Maybe many references in the content were cited wrongly, please check them.

3.      Please indicate the incident angle definition by clear configuration mentioned in the manuscript.

4.      On page 6, Line 190, “Tale 1” should be “Table 1”.

5.      What is the applied shock wave shape during axial overload in simulation in figure 6?

6.      The author used 130000g with 10us impact to the inertial switch, a response speed of 8us was achieved. It seems that the response time is related with the pulse width of applied shock wave, the relationship between them should be investigated.

7.      On page 10, there are two Figure 9, please correct them, and the related explanations are also obfuscated.

8.      In the first figure 9, why the U-shape proof-mass cannot return its initial position once the rotary speed decrease to zero?

9.      The boundary conditions should be given in the dynamic simulation using LS-DYNA.

10.   On page 12, Line 300, how was the switch manually deformed to the “ON” state before testing?

N/A

Round 2

Reviewer 1 Report

Dear authors,

many thanks for your efforts. I think it's fine now.

Sincerely,

Reviewer 2 Report

The authors have addressed all comments, and I think this paper could be accepted.